# Surgical Hand Gesture Recognition Utilizing Electroencephalogram as Input to the Machine Learning and Network Neuroscience Algorithms

**DOI:** 10.3390/s21051733

**Published:** 2021-03-03

**Authors:** Somayeh B. Shafiei, Mohammad Durrani, Zhe Jing, Michael Mostowy, Philippa Doherty, Ahmed A. Hussein, Ahmed S. Elsayed, Umar Iqbal, Khurshid Guru

**Affiliations:** 1Applied Technology Laboratory for Advanced Surgery (ATLAS), Roswell Park Comprehensive Cancer Center, Buffalo, NY 14203, USA; Somayeh.BesharatShafiei@RoswellPark.org (S.B.S.); mdurrani20@gmail.com (M.D.); zhe.jing@roswellpark.org (Z.J.); mgmostowy@gmail.com (M.M.); Philippa.Doherty@RoswellPark.org (P.D.); Ahmed.Aly@RoswellPark.org (A.A.H.); Ahmed.Elsayed@RoswellPark.org (A.S.E.); umar.iqbal@roswellpark.org (U.I.); 2Roswell Park Comprehensive Cancer Center, Department of Urology, Buffalo, NY 14203, USA

**Keywords:** robot-assisted surgery (RAS), electroencephalogram (EEG), functional brain network, surgical gesture detection

## Abstract

Surgical gestures detection can provide targeted, automated surgical skill assessment and feedback during surgical training for robot-assisted surgery (RAS). Several sources including surgical videos, robot tool kinematics, and an electromyogram (EMG) have been proposed to reach this goal. We aimed to extract features from electroencephalogram (EEG) data and use them in machine learning algorithms to classify robot-assisted surgical gestures. EEG was collected from five RAS surgeons with varying experience while performing 34 robot-assisted radical prostatectomies over the course of three years. Eight dominant hand and six non-dominant hand gesture types were extracted and synchronized with associated EEG data. Network neuroscience algorithms were utilized to extract functional brain network and power spectral density features. Sixty extracted features were used as input to machine learning algorithms to classify gesture types. The analysis of variance (ANOVA) F-value statistical method was used for feature selection and 10-fold cross-validation was used to validate the proposed method. The proposed feature set used in the extra trees (ET) algorithm classified eight gesture types performed by the dominant hand of five RAS surgeons with an accuracy of 90%, precision: 90%, sensitivity: 88%, and also classified six gesture types performed by the non-dominant hand with an accuracy of 93%, precision: 94%, sensitivity: 94%.

## 1. Introduction

### 1.1. Importance of Gesture Detection in Robot-Assisted Surgery (RAS)

RAS offers advantages such as three-dimensionality of the surgical field, magnified images of the work area, and improved dexterity compared to the traditional surgical framework. Moreover, RAS has benefits for patients including smaller incisions, decreased blood loss, shorter hospital stays, faster return to work, improved cosmesis, and lower incidence of some surgical complications [1]. While RAS advantages are appreciated, challenges in skill evaluation limits utilization of the robot-assisted technologies, particularly for complex surgical procedures. Existing methods for evaluation of RAS expertise level for trainees and surgeons rely upon subjective, peer-based observational assessment [2], and outcome-based analysis [3]. Such evaluations require large amounts of expert monitoring and manual rating and can be inconsistent due to biases in human interpretation [4]. Previous work on skill evaluation in RAS mainly exploited kinematic data recorded by the robot and global measurements of the task. These measurements include time to completion [5,6], speed and number of hand movements [5], distance travelled [6], and force and torque signatures [6,7,8]. These methods are easy to implement. However, they perform a global assessment about skill level and neglect the fact that a surgical task is composed of several different *gestures*. These skill evaluation methods have two main shortcomings: First, they use a single model for a whole complex task, while segmenting a task into gestures will allow for the use of a simpler model for each gesture. Second, those methods assume that a trainee is either skilled or unskilled at performing a whole task, while a trainee may be skilled in performing some segments of the task and unskilled in performing other segments as a complexity level is different for a gesture.

### 1.2. Literature Review of Gesture Detection in RAS Application

Automated recognition of “gestures” can provide a more accurate method for automated evaluation of surgical performance. Different sources of data including video [9], a robot’s kinematics data [10,11,12,13], and electromyograms (EMG) [14] have been proposed for gesture segmentation and recognition. Lee et al., proposed the deep neural network and leap motion for hand gesture recognition. They used 903 training dataset and 100 testing dataset, to train the proposed network using five types of surgical hand gesture including hovering, grab, click, one peak, and two peaks performed by 10 subjects. They achieved a classification accuracy of 86.46% [15]. In another study, Sarikaya et al., developed a long short-term memory network (LSTM) model that jointly learns temporal dynamics on rich representations of visual and motion features (JIGSAWS dataset), and simultaneously classifies activities of low-level gestures and surgical tasks. The authors trained their model on a fixed random set of 121,200 video segments and used 422 video segments for testing. Average precision of their model was 51% for 3 tasks and 14 possible gesture labels [16].

In addition to variety of modalities used for gesture detection, several algorithms have also been proposed to address this challenge. These algorithms include support-vector-machine (SVM), hidden-Markov models [17,18], and neural networks [15,19,20]. We compared the result of some studies about gesture detection in Table 1.

### 1.3. Strengths and Shortcomings of the Existing Methods of Gesture Detection in RAS Application

While videos contain semantic information that are not presented in kinematic data, they are not typically used because of the complications associated with automatic video interpretation [9]. Instead, recording kinematic data requires additional recording devices. Also, a surgeon’s hand kinematics may not be accessible for recording in the operating room (OR). Moreover, processing surgeon’s hand kinematics, hand trajectory segmentation is another challenge that has not yet been addressed [12].

### 1.4. The Purpose of This Study

While electroencephalogram (EEG) data are source of information about motor, cognition, and perception functions [12,26,27,28,29], this modality has not been widely used for surgical gesture detection. In this study, we proposed functional brain network and power features to be used in machine learning algorithms to classify surgical gestures performed in the operating room (OR) for dominant (eight gesture types) and non-dominant hands (six gesture types). Functional brain network features were extracted by applying network neuroscience algorithms to EEG data. Power features were extracted by applying the short fast Fourier transform (SFFT) method to EEG data. The k-nearest neighbors (KNN), bagged decision trees (BAG), random forest (RF), and extra trees (ET) machine learning models were examined to find the best classification model.

### 1.5. Contribution of This Study

The proposed method will aid in developing an objective model for detection of RAS surgical gestures. Since EEG data recording does not interfere with the main task of surgery and processing EEG data is not as complicated as video processing, the proposed gesture classification method will be useful in addressing this challenge in RAS.

Materials and methods used in this study are explained in Section 2, followed by results in Section 3. The findings of this study are discussed in Section 4.

## 2. Materials and Methods

The study was conducted in accordance with relevant guidelines and regulations and was approved by Roswell Park Comprehensive Cancer Center Institutional Review Board (IRB: I-241913).

### 2.1. Data Recording Setup

A group of five RAS surgeons (dominant hand: right), one master surgeon and four surgical fellows, from Roswell Park Comprehensive Cancer Center (RPCCC) performed *34* robot-assisted radical prostatectomies. The details of the study and the goal of the study were explained to all surgeons and an informed consent was obtained before the first session of study. Brain activity was recorded by placing a 128 EEG headset (ANT neuro inspiring technology, Inc., Hengelo, Netherlands) on the scalp while simultaneously recording the surgical video feed. EEG was recorded at a frequency of 500 Hz using 119 electrodes from frontal (2 channels), prefrontal (3 channels), central (7 channels), temporal (2 channels), parietal (10 channels), occipital (4 channels), frontal-central (19 channels), frontal-temporal (10 channels), parieto-occipital (17 channels), temporal-parietal (8 channels), and central-parietal (18 channels) cortices. From the other nine channels, two are reference electrodes placed on the mastoids, and 7 electrodes (I1, Iz, I2, CPz, PO5, PO6, Oz) were excluded from this study due to lack of signal quality.

### 2.2. Gestures Extraction

Surgical gestures, performed by dominant and non-dominant hands, were extracted individually and synchronized with associated EEG data, by examining the videos of the surgical scenes (Figure 1). Different types of extracted gestures (totally 1024 gestures performed by left hand and 1258 by right hand) are represented in Table 2. These gestures include:Bipolar Cautery: Surgeon uses bipolar tool to control bleeding event by cauterization. While cauterization is passing, a high frequency electrical current through tissue from one electrode to another.Monopolar Cautery: Surgeon uses regular monopolar tool to control bleeding by cauterization.Blunt Dissection: Surgeon separates tissue planes by pushing them around, instead of cutting them or cauterization.Tissue Grasping: Surgeon catches tissue.Retraction: Surgeon retracts structures by holding them aside, to improve operative field visibility.Suturing: Surgeon uses surgical sutures to hold body tissues together. Sutures (or stitches) are typically applied using a needle with an attached piece of thread and are secured with surgical knots.Needle Insertion: Surgeon inserts the needle to the entry site of tissue surface using downward pressure and applies a twisting motion until resistance decreases as the needle passes through the surface and the suture begins to traverse the tissue.Surgical Thread Grasping: Surgeon grasps the surgical thread.Idle: Surgeon is not doing any action as it may be waiting for a response from the surgical team or is thinking before deciding the next step.

### 2.3. Electroencephalogram (EEG) Data Analyses

We used the Advanced Source Analysis (ASA) framework developed by ANT neuro inspiring technology, Inc, Netherlands, to pre-process EEG data. ASA incorporates artifact correction by spatial filtering. It separates brain signal from artifacts based on their topography and subsequently removes the artifacts without distorting the brain signal. The separation is determined based on data intervals with a clear artifactual activity as selected by the user and will be used to specify the artifact topography. The method determines which part of the data is considered the brain signal using two criteria. The first criterion specifies the highest permitted amplitude of the brain signal while the second criterion specifies the highest correlation between brain signal and artifact topography permitted. Then, a spatial principal component analysis (PCA) method is used to determine the topographies of the artifact-free brain signals and the artifact signals. Finally, the artifact components are removed. It should be mentioned that in the EEG recording system, an active shielding technique protects the referential EEG inputs from environmental noise (e.g., grid interference noise and cable movement). Also, by using the EEGO framework for EEG recording, a running DC offset value was calculated per channel over the data. This offset was subtracted from the data to compensate for the DC offset. Artifacts by line noise were removed by applying a 60 Hz notch filter to EEG data. The EEG data from channels were filtered with a band-pass filter (0.2–250 Hz) with a filter steepness of 24 dB/octave. The EEG artifact correction was done based on blind source separation and using the topographical PCA-based method. Individual portions of EEG were visually inspected for facial and muscular activity artifacts and other artifacts [30]. Then, the spatial Laplacian (SP) technique was applied to the signals and the result was used for extracting features [31].

### 2.4. Types of Feature and Functional Brain Network Analyses

We extracted two groups of features; functional brain network and power spectral density (PSD) features. We also, considered motor (included 70 channels), cognition (included 37 channels), and perception (included 12 channels) cortices in our analyses. To be able to extract functional brain network features, we extracted functional brain network by using EEG data.

Brain network: Our approach was to consider EEG channels as nodes (i; i = 1, ….,*N*, where *N* = 119) of the functional brain network and strength of functional connectivity (FC) between pairs of brain regions (EEG channels) as the link between nodes of the functional brain network. Functional connectivity was estimated as the coherence of the associated EEG channels’ time series and can be considered as a measure of communication between brain regions [32,33]. The functional brain network will then be a square matrix whose elements (*i,j*) represent the magnitude of FC between pairs of brain areas. The result is a weighted connectivity matrix ($\Gamma \;=\;\left(\Gamma_{\mathrm{ij}}\right)\in {ℜ}^{\mathrm{NXN}}$) whose entries represent the connection weight between different brain areas i and j (EEG channels) and were also specific to each individual. Since, this network fluctuates over different timescales; it is possible to study time-varying properties of the FC network and find the relationship between these time-varying properties and associated external or internal cause/stimulation. Community structure, the decomposition of a network into densely inter-connected sub-networks or “communities”, is one method for analyzing dynamic features of FC networks [34,35,36]. Communities representing brain regions tend to preferentially communicate with each other, while weakly communicating to the rest of the brain regions [37,38,39].Brain functional modules: The adjacency matrix (Γ_ij_) of each 1-sec window within the recording was used in categorical multilayer community detection algorithm [40]. A Louvain-like locally “greedy” algorithm was used to optimize modularity in the multilayer network [41,42]. Since the community detection algorithm is non-deterministic [43], and also due to near-degeneracies in the modularity optimization [44], the output typically varies from one run to another. Optimization of the multi-layer modularity was repeated 100 times, in a consensus iterative algorithm for each single run [43]. This was done to identify a single representative partition from all partition sets, based on statistical testing in comparison to the ‘Newman–Girvan’ null network [42]. The output of categorical multilayer community detection is a partition matrix (A), representing the functional community that each channel was assigned to [40].Module-allegiance matrix: The module allegiance matrix (MAM) was derived using the functional community each channel was assigned to. MAM elements represent the probability that pairs of brain areas be assigned to the same functional community during processing tasks. This matrix was used to extract dynamic brain features of network flexibility, integration and recruitment.

Functional brain network features were extracted by using:-Adjacency matrices (Γ; extracted by coherence analysis applied to EEG data): search information, strength, transitivity, mean pairwise diffusion efficiency, global efficiency, and mean global diffusion efficiency features were extracted by using Γ.-Module-allegiance matrix (MAM; extracted by applying multi-layer community detection techniques to adjacency matrices): integration and recruitment features were extracted by using MAM.-Partition matrices (A; extracted by applying community detection techniques to adjacency matrices): regional network flexibility feature was extracted using A.

PSD features were extracted by applying SFFT to EEG data.

### 2.5. Definition of Extracted Features

#### 2.5.1. Brain Regional (Node) Network Flexibility

The flexibility of each node of a network corresponds to the number of times that it changes module allegiance [45]. We used partition matrix A to calculate flexibility of network nodes (regional flexibility) as (1) [46]:(1)$$f_{i}=1-\frac{1}{T-1}\overset{T-1}{\underset{r=1}{\sum }}\delta \left(A_{i,r},A_{i,r+1}\right)$$
where, ‘i’ is the EEG channel, A is partition matrix, and r is the time layer (successive one second windows). Regional network flexibility of region ‘i’ counts the portion of times that brain region ‘i’ changes its community assignment in successive one second windows throughout a gesture performance [45]. Low (high) flexibility score shows that the corresponding brain region’s community assignment is consistent (variable) across layers [45,47].

#### 2.5.2. Integration

Average probability that a brain area is in the same network community as areas from other cortices [46,48].

#### 2.5.3. Recruitment

Average probability that a brain area is in the same network community as other areas from its own cortex [48].

Extracted MAM matrices were used to evaluate integration and recruitment of channels within each cortex. The average value of each metric—integration, and recruitment—over all channels in each cortex was considered flexibility, integration, and recruitment of that cortex. We also evaluated these features for cortices in left and right lobes of the brain, separately.

#### 2.5.4. Search Information

The amount of information (measured in bits) that is required to follow the shortest path between a given pair of nodes [49]. The adjacency matrix of each recording was used to extract this feature [50]. Search information was calculated for pairs of channels within each cortex, and the average value was considered per cortex in each recording. This feature was evaluated for all cortices in the left and right lobes and throughout the cortices of the whole brain.

#### 2.5.5. Strength

The total communication weight of channels within each cortex of the brain. The adjacency Matrix of each recording was used to extract the strength feature. The average strength for each cortex was considered as strength of that cortex [51]. Average value for cortices in the left and right lobes of the brain were evaluated separately as the strength of cortices in the left and right lobes. This feature was evaluated for all cortices in the left and right lobes and throughout cortices of whole brain.

#### 2.5.6. Mean Pairwise Diffusion Efficiency of Cortices

The diffusion efficiency between nodes ‘i’ and ‘j’ is the inverse of the mean first passage time from ‘i’ to ‘j’, that is the expected number of steps it takes a random walker starting at node i to arrive for the first time at node j [52]. The average value of pairwise diffusion efficiency for channels in each cortex is considered as mean global diffusion efficiency of that cortex. The value of this feature was calculated separately for cortices in the left and right lobes of the brain.

#### 2.5.7. Transitivity

Transitivity is calculated as the ratio between the observed number of closed triplets and the maximum possible number of closed triplets in the network. This feature shows overall probability for the network to have adjacent nodes interconnected, thus revealing the existence of tightly connected subgroups [53].

#### 2.5.8. Global Efficiency

The average of inverse shortest path length. Efficiency was computed using an auxiliary connection-length matrix L, defined as Lij = 1/Aij for all nonzero Lij; This has an intuitive interpretation, as higher connection weights intuitively correspond to shorter lengths [54,55].

#### 2.5.9. Power

The SFFT method with a one second Kaiser moving window was used to calculate the PSD of EEG signals. A 50% overlap was considered for Kaiser moving window. The PSD analysis was used to extract average power of EEG signals of channels within each brain cortex (motor, cognition, and perception).

A summary of extracted features is shown in Table 3.

### 2.6. Machine Learning Algorithms

Gestures from dominant and non-dominant hand include 8 and 6 types, respectively. Available data for different classes are imbalanced, meaning the number of available data samples was different for various gesture types. Most machine learning algorithms assume that all classes have an equal number of samples. Therefore, we used an oversampling technique of the synthetic minority oversampling technique (SMOTE), to synthesize new examples of the minority types so that the number of examples in the minority class better resembles or matches the number of examples in the majority classes [56]. SMOTE was applied only to training sets, not the entire dataset.

We then implemented different non-linear machine learning classifier algorithms, frequently used in EEG classification, to data in order to find the best classifier. We evaluated the following machine learning models of KNN, BAG, RF, and ET, on the datasets for the dominant and non-dominant hands.

KNN: The KNN is a very simple technique. The entire training dataset is stored. When a prediction is required, the k-most similar records to a new record from the training dataset are then located. From these neighbors, a summarized prediction is made. Similarity between records can be measured in many ways. Once the neighbors are discovered, the summary prediction can be made by returning the most common outcome or taking the average. KNN parameters include number of neighbors to use: 5, weight function used in prediction: ‘uniform’.Bagging: Bagging involves taking multiple samples from training dataset (with replacement) and training a model for each sample. The final output prediction is averaged across the predictions of all the sub-models. The three bagging models are BAG, RF, and ET. Bagging performs best with algorithms that have high variance.RF: Random forest is an extension of bagged decision trees. Samples of the training dataset are taken with replacement, but the trees are constructed in a way that reduces the correlation between individual classifiers. Specifically, rather than greedily choosing the best split point in the construction of the tree, only a random subset of features is considered for each split.ET: Extra trees are another modification of bagging where random tree are constructed from samples of the training dataset. The ET algorithm works by creating a large number of unpruned decision trees from the training dataset. Classification predictions are made by using majority voting.

The ER and RF algorithms are very similar ensemble methods as both are composed of many decision trees, where the final decision is obtained considering the prediction of every tree. Also, when selecting the partition of each node, both randomly choose a subset of features [57]. However, RF and ET have some differences including:-RF uses bootstrap replicas: it subsamples the input data with replacement, whereas ET uses the whole original sample.-The selection of the cut points in order to split nodes: RF chooses the optimal split while ET chooses it randomly. However, after selection of the split points, the two algorithms choose the best between all the subset of features. Therefore, ET adds randomization but still has optimization.

Bias-variance tradeoff: the differences between RF and ET motivate the reduction of both bias and variance. While using the whole original sample instead of a bootstrap replica will reduce bias, choosing randomly the split point of each node will reduce variance [57].

The BAG, RF, and ET parameters that we used in our analyses had: 350 trees; the function to measure the quality of split (criterion): Gini impurity; the minimum number of samples required to split an internal node: 2; the minimum number of samples required to be at a leaf node: 1.

We examined a range of number of trees in the ensemble algorithms, from 100 to 1000 with increment of 20, to acquire the best performance. We set number of trees to 350. All models are implemented in Scikit-learn library, Python 3.7.

### 2.7. Feature Selection Method

Feature selection is a process of automatically selecting features that contribute most to the output. Statistical tests can be used to select those features that have the strongest relationship with the output variable. We used analysis of variance (ANOVA) F-value statistical method for feature selection.

### 2.8. Measures of Classification Method’s Performance

The 10-fold cross-validation was used to validate gesture-type classification method and to investigate the effect of the training dataset size on the classification performance [57].

Classifications were evaluated regarding common metrics of average accuracy, precision, and sensitivity. Average accuracy: the ratio between the sum of correct predictions and the total number of samples; Precision is the ratio of correct positive predictions (Tp) and the total positive results predicted by the classifier (Tp + Fp) and sensitivity represents the ratio of positive predictions (Tp) and the total positive results (Tp + Fn).

Also, we applied paired *t*-test (α = 0.05) with Bonferroni correction to extracted accuracies for different number of selected features, to confirm whether detected differences among the results of BAG, RF, and ET machine learning methods are statistically significant.

## 3. Results

Classification results for different machine learning methods and different number of selected features are represented in Figure 2 for dominant and non-dominant hands.

Our results showed that the best classification accuracy occurs with 60 features (57 functional brain network and 3 PSD features) and ET classifier. For dominant hand (eight gestures), classification accuracy: 90.2%, precision: 89.81%, sensitivity: 88.33%. For non-dominant hand (six gestures) classification accuracy: 93.4%, precision: 94.00%, sensitivity: 94.23%.

By increasing number of selected features, classification accuracy improves for both hands’ gestures. Also, bagging models (BAG, RF, and ET) work better than the KNN machine learning model for this classification problem.

We found that BAG, RF, and ET accuracies of classifying gesture types performed by dominant and non-dominant hands are significantly different (*p* < 0.0001). The level of accuracy improvement for pairs of classification methods is represented in Table 4.

Also, results of applying classification methods to 60 features are shown in Table 5, for gestures performed by dominant and non-dominant hands.

To extract confusion matrix for classifying types of gestures performed by dominant (8 types) and non-dominant (6 types) hands using ET algorithm, we considered 75% of data as training dataset and 25% of data as test set. Confusion matrices are represented in Table 6 and Table 7 for dominant and non-dominant hands, respectively.

## 4. Discussion

Automatic detection of hand gestures in RAS is required for evaluating surgical skills and providing surgical trainees with structured feedback [58]. Robot end effector kinematics and surgical videos are the most common modalities proposed for RAS gesture detection. However, recording end effector kinematics may not be approved in the operating room. While using videos does not interfere with a surgeon’s performance, processing videos is costly. Also, RAS gesture detection models developed by using videos are mostly complicated deep neural network models. Moreover, the developed method depends on the surgery type and is not generally used for all types of surgery. High complexity level and cost may limit using videos for gesture detection. Analyzing robot end-effector kinematics is less complicated. However, recording robot end effector kinematics may be feasible only in research labs and during practice on surgical simulators as attaching an external tracking sensor to surgical robot in the OR is not possible.

### 4.1. Proposed Method and Implications

To address this challenge, we proposed utilizing functional brain network features extracted from EEG data in combination with PSD features in ET classifier. EEG data were recorded from five RAS surgeons performing RARP. The surgical videos, synchronized with EEG data, were used to extract bipolar cautery, monopolar cautery, blunt dissection, tissue grasping, retraction, suturing, needle insertion, surgical thread grasping, and idle gestures performed by dominant and non-dominant hands and also extract their corresponding EEG data. Implementing network neuroscience and community detection algorithms, we extracted functional brain network features including regional network flexibility, integration, recruitment, search information, strength, transitivity, mean pairwise diffusion efficiency, and global efficiency. These features and PSD feature were used in the gesture classification process. We have shown that functional brain network and PSD features were informative for dominant and non-dominant hands’ gesture classification in RAS application. We achieved 90.2% and 93.4% accuracy in classifying gestures performed by the dominant hand (8 gestures) and non-dominant hand (6 gestures), respectively. The extracted accuracies illustrate the importance of brain dynamic measurements in understanding information about surgical gestures.

### 4.2. Strength of Proposed Method

While existing RAS gesture classification methods have their own advantages and applications, our proposed method demonstrates higher classification accuracy and feasibility of differentiating between gestures performed by dominant and non-dominant hands simultaneously. In RAS, handedness (the better, faster, or more precise performance or individual preference for use of a hand) is a required critical skill [59,60,61]. RAS surgeons should be able to manage use of dominant and non-dominant hands for executing different gestures in an efficient way. It has been shown that hand movement trajectory smoothness is higher for expert surgeons compared to novices and intermediates. Also, smoothness is higher for non-dominant hand in comparison with dominant hand [13]. Hence, it is important to discriminate between gesture detection performed by each individual hand of the surgeon. Our understanding is that this is the first study taking this fact into consideration.

Functional brain network reconfigures by practice throughout skill acquisition [62]. Although subjects have different levels of expertise, all of them had required skills to perform considered gestures on the patient, with high performance and under the supervision of a master surgeon in the OR. Therefore, differences in evaluated functional brain network features are not subject to change as a result of skill acquisition.

Inter- and intra-subject differences were considered in the proposed method, as we included EEG data from five RAS surgeons with varying expertise levels (fellows and expert surgeon). Moreover, EEG data have high spatial and temporal resolution as EEG was recorded from 119 areas of the brain at frequency of 500 Hz.

We also examined different machine learning algorithms for classification including KNN, BAG, RF, and ET classifiers. Our results showed that the ET algorithm performs better on our data. The proposed gesture classification method has several advantages for researching the RAS environment as it is convenient and does not interfere with a surgeon’s performance. Also, analyzing EEG data is not as complicated as video processing.

Objective gesture recognition for dominant and non-dominant hands can provide a valuable method for automated evaluation of surgical performance. Recognizing type of gesture performed by dominant and non-dominant hands of RAS surgeon throughout a complicated surgical task will aid in evaluating the skill level of surgeons in performing each individual gesture. This detailed and fair skill evaluation will help RAS trainees to focus only on performing gestures where their skill level is low. These improvements in skill evaluation will result in accurate feedback to trainees and consequently shortens learning curve.

### 4.3. Limitations and Shortcomings of the Proposed Method

Data from only one master surgeon and four surgical fellows were used in this study. To validate the wide-ranging application of proposed method, data from more experts and also data from novice and beginner RAS trainees are required.

To the best of our knowledge, this is the first study proposing brain functional network features for classifying RAS surgical gestures performed by dominant and non-dominant hands of surgeons. Using EEG data in network neuroscience algorithms to extract informative features for surgical gesture detection can address some of the existing challenges. However, more in-depth investigations are required to explore other informative features of EEG data and more validations are required to develop a general automatic gesture detection model for surgical application in a clinical framework.

## 5. Conclusions

Based on our classification results with different numbers of selected features (up to 60), by increasing number of selected features classification accuracy improves. It shows that the proposed and developed 60 features are informative in surgical gesture recognition, but there are still informative features to be added to this feature set to make classification accuracy even better. We will consider extracting features through distinct frequency domains and individual brain cortices and using them for RAS gesture recognition, as a future goal.

The developed gesture recognition method uses the EEG data and these data are not dependent on the type of surgery. Also, the considered gestures are common in different types of surgery. Therefore, the developed method can be generalized to other types of surgery by adding more gesture types from other types of surgery, as a future goal.

Since a command for performing any action is created and controlled by the brain, utilizing EEG data of RAS surgeons performing surgical tasks can be used not only for dominant and non-dominant gesture detection, but also to identify the decision making period for each gesture command creation. It may also be useful to extract the relationship between length of decision-making time window and the surgeon’s performance.

## Figures and Tables

**Figure 1 sensors-21-01733-f001:**
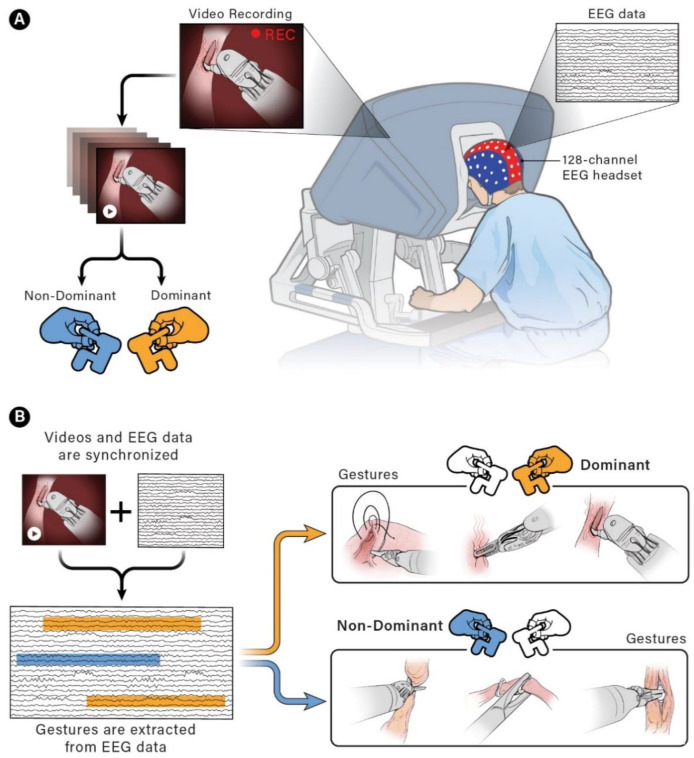
Schematic of experimental set up and extraction of electroencephalogram (EEG) data associated with each gesture. (**A**) EEG and video data recording set up. (**B**) Synchronizing EEG data with videos to extract EEG portion associated with each gesture performed by dominant and non-dominant hands.

**Figure 2 sensors-21-01733-f002:**
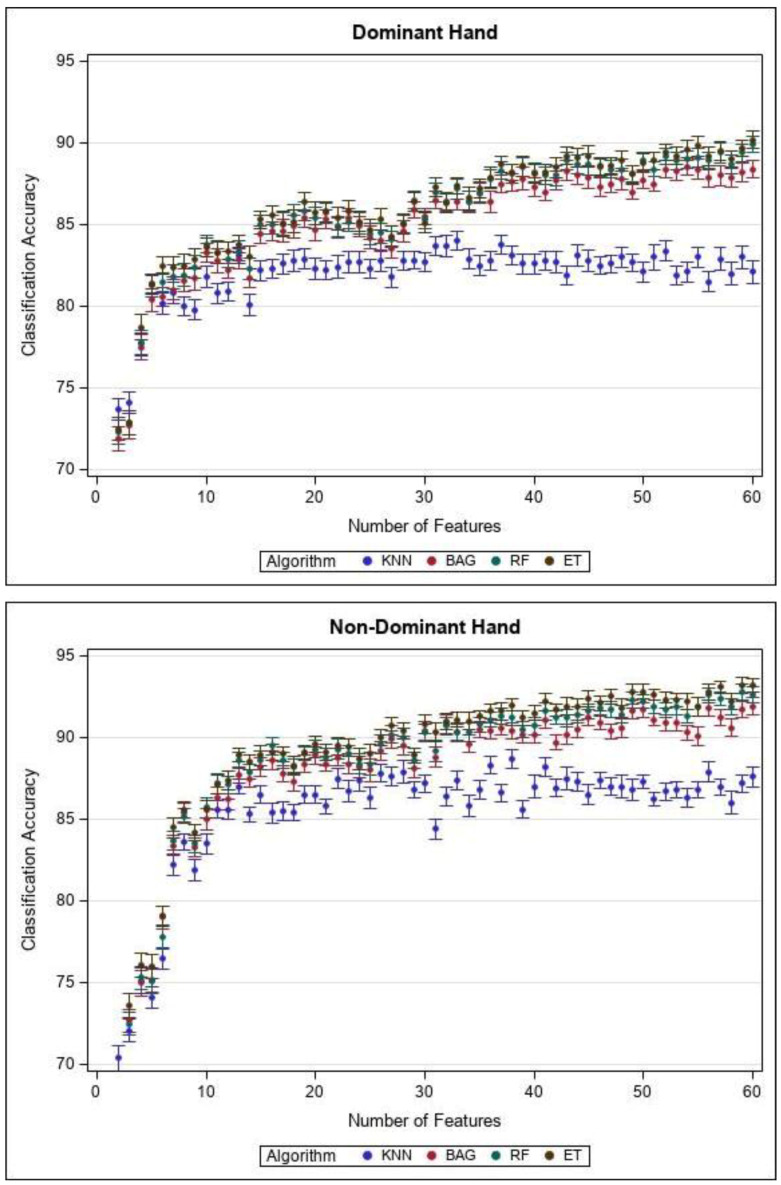
Accuracy and standard error for different machine learning methods and different number of selected features for dominant and non-dominant hands.

**Table 1 sensors-21-01733-t001:** Comparing our proposed method with other robot-assisted surgery (RAS) gesture classification methods.

Study	Data	Classification Method	Task	# of Classes	Accuracy	Distinct Hands?
Henry C. Lin, et al. [21]	End effector motion data (JIGSAWS [22])	linear discriminant analysis	Robot-assisted surgery tasks performed on simulator	8 gestures	91.52%	No
Xiaojie Gao, et al. [23]	End effector motion data (JIGSAWS [22])	reinforcement learning and tree search	Robot-assisted surgery tasks performed on simulator	8 gestures	81.67%	No
Fabien Despinoy [10]	End effector motion data	k-nearest neighbors	pick-and-place performed using Raven-II robot	12 gestures	81.9%	No
Duygu Sarikaya et al. [16]	Surgical videos (JIGSAWS [22])	long short-term memory network (LSTM)	Robot-assisted surgery tasks performed on simulator	14 gestures	51%	No
	Surgical videos (JIGSAWS [22])	optical flow ConvNets		3 tasks	84.36%	No
Duygu Sarikaya and Pierre Jannin [24]	Suturing task of the JIGSAWS [22]	spatial temporal graph convolutional networks	Robot-assisted surgery tasks performed on simulator	10 gestures	68%	No
Francisco Luongo et al. [25]	videos of a live vesico-urethral anastomos (VUA)	LSTM and convLSTM)	Robot-assisted surgery tasks	5 suturing gestures	87%	No

**Table 2 sensors-21-01733-t002:** Extracted gestures performed by dominant and non-dominant hands.

Gesture Schematic	Gesture Name	Explanation	Hand
** 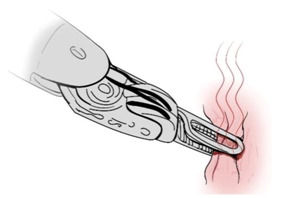 **	Bipolar Cautery	The use of cautery to control bleeding using bipolar device.	Dominant
** 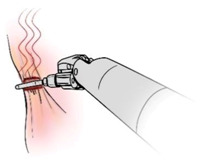 **	Monopolar Cautery	The use of cautery to control bleeding using monopolar (regular) device.	Dominant
** 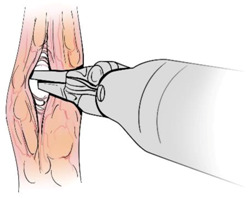 **	Blunt Dissection	Separating tissue planes by “pushing” rather than cutting or cautery.	Dominant
** 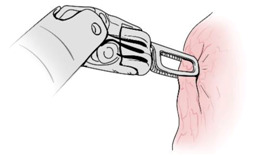 **	Tissue Grasping	To catch tissue.	Non-dominant
** 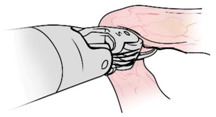 **	Retraction	Retraction by holding structures aside to improve visibility of the operative field.	Dominant, non-dominant
** 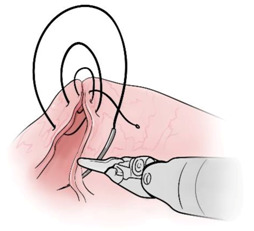 **	Suturing	The use of surgical sutures to approximate edges including knot tying.	Dominant, non-dominant
** 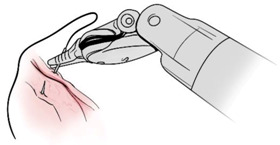 **	Needle Insertion	Initial contact and passing of the needle through the tissue up to the point when suture begins to traverse the tissue.	Dominant, non-dominant
** 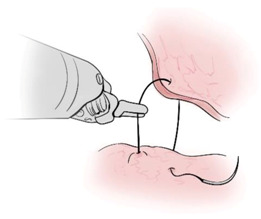 **	Surgical Thread Grasping	To catch surgical thread.	Dominant, non-dominant
** 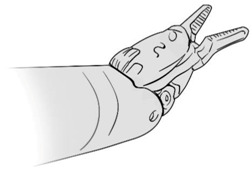 **	Idle	Not doing anything.	Dominant, non-dominant

**Table 3 sensors-21-01733-t003:** Summary of extracted features.

Feature	Method of Extraction		Total Number
Regional network flexibility	Network community detection technique applied to functional connectivity matrix	Calculated for motor, cognition, and perception cortices at left and right lobes of the brain and same cortices throughout brain	9
Integration			9
Recruitment			9
Search Information	Functional connectivity matrix		9
Strength			9
Mean pairwise diffusion efficiency of cortices			9
Transitivity		Calculated throughout brain	1
Global efficiency			1
mean global diffusion efficiency			1
Power	Short fast Fourier transform (SFFT)	Calculated for motor, cognition, and perception cortices	3

**Table 4 sensors-21-01733-t004:** Results of applying paired *t*-test (α = 0.05) with Bonferroni correction to extracted accuracies using bagged decision trees (BAG), random forest (RF), and extra trees (ET) algorithms (D: dominant hand, ND: non-dominant hand).

Test	Improvement	Confidence Interval	*p*-Value
**D; ET versus BAG**	0.90%	0.78–1.02%	<0.0001
**D; RF versus BAG**	0.62%	0.53–0.72%	<0.0001
**D; ET versus RF**	0.28%	0.21–0.36%	<0.0001
**ND; ET versus BAG**	1.07%	0.95–1.18%	<0.0001
**ND; RF versus BAG**	0.58%	0.45–0.70%	<0.0001
**ND; ET versus RF**	0.49%	0.41–0.57%	<0.0001

**Table 5 sensors-21-01733-t005:** Classification result—accuracy (standard deviation)—for dominant (D) and non-dominant (ND) hands gestures, using 60 features and different non-linear classifiers including k-nearest neighbors (KNN), bagged decision trees (BAG), random forest (RF), and extra trees (ET) classifiers.

	KNN	BAG	RF	ET
D	82.8 (0.023)	88.9 (0.017)	89.8 (0.017)	**90.2 (0.018)**
ND	86.7 (0.023)	91.9 (0.015)	92.7 (0.016)	**93.4 (0.017)**

**Table 6 sensors-21-01733-t006:** Confusion matrix for classification of eight surgical gestures performed by dominant hand.

Dominant Hand		Predicted Label							
		Bipolar Cautery	Monopolar Cautery	Blunt Dissection	Retraction	Suturing	Needle Insertion	Surgical Thread Grasping	Idle
**True Label**	Bipolar Cautery	83%	0%	0%	7%	1%	0%	0%	9%
	Monopolar Cautery	0%	100%	0%	0%	0%	0%	0%	0%
	Blunt Dissection	0%	0%	99%	0%	1%	0%	0%	0%
	Retraction	9%	0%	0%	84%	0%	0%	0%	7%
	Suturing	0%	0%	0%	0%	94%	3%	0%	3%
	Needle Insertion	0%	0%	0%	0%	3%	92%	1%	4%
	Surgical Thread Grasping	0%	0%	0%	0%	1%	0%	99%	0%
	Idle	23%	0%	6%	4%	0%	4%	0%	63%

**Table 7 sensors-21-01733-t007:** Confusion matrix for classification of six surgical gestures performed by non-dominant hand.

Non-Dominant Hand		Predicted Label					
		Tissue Grasping	Retraction	Suturing	Needle Insertion	Surgical Thread Grasping	Idle
**True Label**	Tissue Grasping	95%	5%	0%	0%	0%	0%
	Retraction	2%	86%	0%	0%	5%	7%
	Suturing	0%	0%	100%	0%	0%	0%
	Needle Insertion	0%	0%	0%	99%	1%	0%
	Surgical Thread Grasping	1%	0%	0%	0%	99%	1%
	Idle	0%	15%	0%	2%	5%	78%

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
