# Peer review of "Surgical Hand Gesture Recognition Utilizing Electroencephalogram as Input to the Machine Learning and Network Neuroscience Algorithms"

_sensors, 2021, doi:10.3390/s21051733_

Round 1

Reviewer 1 Report

This paper introduces a method to extract features from EEG and used machine learning algorithms to distinguish gesture types during surgery. 

Authors should indicate the manufacturer of the net used and sampling frequency in materials and methods.

Intensive care units/operation theaters are prone to line noise; hope authors took care of subharmonics and harmonics of line noise in addition to notch filter at 60 Hz.

The methods of the paper are not very clear about using PCA and ICA. If artifacts were corrected using PCA, what is the point of using ICA, and what is the meaning of the most significant ICs? What is the range of ICs used across subjects?

EEG amplitude is generally in microvolts. The authors used an absolute threshold of +/- 75 mv, which seems very large. When the word absolute is used, no need of specifying +/-.

A 128-channel EEG headset was used in the study. The number of nodes was 119 out of 128. What was the rationale for removing nine electrodes?

What was the rationale for choosing SVM, KNN, BAG, RG, and ET or is it arbitrary? What are the default model parameters?

Authors should mention the list of features used for 10, 14, 18, 22, 24, 30, 40, 50, and 60? What are the criteria for choosing a random number of features? When using several features, the classification could be better or worse than a smaller number of features. Instead of an arbitrary number of features, there should be a criterion to select features.

In the results, it was mentioned that functional brain network and PSD features gave the highest accuracy. It was not clear how many number of features were used for each.

Author Response

Thank all reviewers for their thoughtful comments. We addressed their comments here and throughout the text of the draft. 

Reviewer 1:
-    Authors should indicate the manufacturer of the net used and sampling frequency in materials and methods.
We added the name of manufacturer and sampling frequency. Page 3, line 109 & 111
-    Intensive care units/operation theaters are prone to line noise; hope authors took care of subharmonics and harmonics of line noise in addition to notch filter at 60 Hz.
We explained all filters and DC offset that were applied to EEG data. Page 7, lines 146-168.
-    The methods of the paper are not very clear about using PCA and ICA. If artifacts were corrected using PCA, what is the point of using ICA, and what is the meaning of the most significant ICs? What is the range of ICs used across subjects?
We updated this section of draft. Page 7, lines 146-168.
-    EEG amplitude is generally in microvolts. The authors used an absolute threshold of +/- 75 mv, which seems very large. When the word absolute is used, no need of specifying +/-.
We updated this section. Page 7, lines 146-168.
-    A 128-channel EEG headset was used in the study. The number of nodes was 119 out of 128. What was the rationale for removing nine electrodes?
We addressed this comment. Page 3, Lines 111-116.
-    What was the rationale for choosing SVM, KNN, BAG, RG, and ET or is it arbitrary? What are the default model parameters?
We addressed this comment. Page 10, lines 283-284. 

-    Authors should mention the list of features used for 10, 14, 18, 22, 24, 30, 40, 50, and 60? What are the criteria for choosing a random number of features? When using several features, the classification could be better or worse than a smaller number of features. Instead of an arbitrary number of features, there should be a criterion to select features.

We updated this section by adding Figure 2. Page 12.  

-    In the results, it was mentioned that functional brain network and PSD features gave the highest accuracy. It was not clear how many number of features were used for each.
We addressed this comment. Page 9, line 271-page 10, line 273 AND page 1  , lines 351-352. 

Reviewer 2 Report

Review for the Manuscript (1058556): Surgical hand gesture recognition utilizing Electroencephalogram as input to the machine learning and network neuroscience algorithms

The paper proposes brain functional network features for classifying RAS surgical gestures performed by dominant and non-dominant hands of surgeons using EEG data in network neuroscience algorithms. The data used to extract meaningful features for surgical gesture detection.  

I have the following comments: 

Page 7 : equation 1 should be labelled and all variables need to be defined.

Features section need to be re-written to fit in a scientific paper as it is talking about each part  separately all parts should be connected to each other’s to encourage the reader to continue reading the paper.  You might also add a block diagram and then you can talk about each part to make it simpler and clearer.

Best Wishes   

Author Response

Reviewer 2:

  • Page 7 : equation 1 should be labelled and all variables need to be defined.

We addressed this comment – Page 9, Line 224

  • Features section need to be re-written to fit in a scientific paper as it is talking about each part separately all parts should be connected to each other’s to encourage the reader to continue reading the paper.  You might also add a block diagram and then you can talk about each part to make it simpler and clearer.

We addressed this comment. Feature section is re-written. Page 8, line 207 to page 10, line 273.

Reviewer 3 Report

Authors aim to present a novel approach to hand gesture recognition using EEG as input to the machine learning and network neuroscience algorithms. The paper is overall well written, and research is interesting. However, some parts of the paper need to be improved.

  1. Introduction should be more focused. Form the introduction in the following manner: 1st paragraph  - the brief overview of the current knowledge on the topic; 2nd paragraph - direction toward the purpose of the paper; 3rd paragraph - the purpose of the paper and it states briefly methodology that has been utilized in the paper; 4th paragraph – explain what is the contribution of the paper, in relation to several previous papers; 5th paragraph - describe other sections of the paper.
  2. Add a new chapter Literature review. The current text in the literature review is relevant, but need to be more structured and expanded with a more in-depth discussion of previous work. It would be useful to divide the chapter into subchapters. 
  3. The chapter on Materials and methods is hard to follow. I suggest that you also divide this chapter into subchapters so that it is easier to read and be more clear. Brain cortices are described in a vague manner. EEG also does not simply catch data on these cortices, since they are distributed in several areas. Machine learning algorithms are described in a too general manner. For each of the algorithms, more information should be provided with regards to other similar research that use them. In addition, it should be discussed on how they tackle sparse data. 
  4. The most relevant algorithm (ET) was described in only one sentence!
  5. Authors should describe why they used 1000 trees. 
  6. More information should be provided about feature selection. What is an initial number of features? Which features were selected for analysis?
  7. Classification results are shown in Table 2. However, statistical tests are needed to confirm if detected differences among the machine learning method are statistically significant. 
  8. The paper presents only the overall classification. However, for the best methods, the full classification matrix should be presented. The issue of classification small subset of goal data should be also discussed in the context of the classification matrix. 
  9. Table 3 should be in the Literature review, but without the last row. 
  10. The discussion indicates that the important feature of this review is that it focuses on distinct hands. There should be more evidence of why this is important. 
  11. Conclusion and discussion should be merged. In the last section, please focus on “Discussion, Implication, and Conclusion” to include
    (1).     Discussion, Implication, and Conclusion
    (2).     Discussion why the authors found out these results and how they comply (or not) with the Literature Review?
    (3).     Conclusions
    (4).     Managerial and Academic Implications
    (5).     Limitations of the paper
    (6).     Future Studies and Recommendations
  12. It seems to me that usage of EEG is not practical. This should be discussed more in the paper and stressed as the limitation of the paper. 

Author Response

Reviewer 3:

  • Introduction should be more focused. Form the introduction in the following manner: 1st paragraph  - the brief overview of the current knowledge on the topic; 2nd paragraph - direction toward the purpose of the paper; 3rd paragraph - the purpose of the paper and it states briefly methodology that has been utilized in the paper; 4th paragraph – explain what is the contribution of the paper, in relation to several previous papers; 5th paragraph - describe other sections of the paper.

We addressed this comment throughout introduction.

  • Add a new chapter Literature review. The current text in the literature review is relevant, but need to be more structured and expanded with a more in-depth discussion of previous work. It would be useful to divide the chapter into subchapters. 

We addressed this comment throughout introduction.

  • The chapter on Materials and methods is hard to follow. I suggest that you also divide this chapter into subchapters so that it is easier to read and be more clear. Brain cortices are described in a vague manner. EEG also does not simply catch data on these cortices, since they are distributed in several areas. Machine learning algorithms are described in a too general manner. For each of the algorithms, more information should be provided with regards to other similar research that use them. In addition, it should be discussed on how they tackle sparse data. 

We updated all parts of materials and methods section. Also, Table 2 is added for more clarification.

  • The most relevant algorithm (ET) was described in only one sentence!

We addressed this comment. Page 10 line 308- page 11 line 325.

  • Authors should describe why they used 1000 trees. 

We addressed this comment page 11, lines 326-327.

  • More information should be provided about feature selection. What is an initial number of features? Which features were selected for analysis?

We addressed this comment. Page 11 line 329-333. Also, Figure 2 is added (page 12).

  • Classification results are shown in Table 2. However, statistical tests are needed to confirm if detected differences among the machine learning method are statistically significant. 

We addressed this comment. Page 11 lines 344-346 AND Page 13, lines 366-373.

  • The paper presents only the overall classification. However, for the best methods, the full classification matrix should be presented. The issue of classification small subset of goal data should be also discussed in the context of the classification matrix. 

We addressed this comment. Page 13 line 381-Page 14 line 391.

  • Table 3 should be in the Literature review, but without the last row. 

We addressed this comment. Page 2 line 71-page 3 line 72.

  • The discussion indicates that the important feature of this review is that it focuses on distinct hands. There should be more evidence of why this is important. 

We addressed this comment. Page 15 lines 426-434.

  • Conclusion and discussion should be merged. In the last section, please focus on “Discussion, Implication, and Conclusion” to include
    (1).     Discussion, Implication, and Conclusion
    (2).     Discussion why the authors found out these results and how they comply (or not) with the Literature Review?
    (3).     Conclusions
    (4).     Managerial and Academic Implications
    (5).     Limitations of the paper
    (6).     Future Studies and Recommendations

We addressed this comment by updating discussion section.

  • It seems to me that usage of EEG is not practical. This should be discussed more in the paper and stressed as the limitation of the paper. 

We addressed this suggestion. Page 15, lines 466-467.

Round 2

Reviewer 3 Report

Dear authors, congratulations on your work, to my opinion the paper can now be accepted.